# Research on Three-Dimensional Porous Composite Nano-Assembled α-MnO_2_/Reduced Graphene Oxides and Their Super-Capacitive Performance

**DOI:** 10.3390/ma15238406

**Published:** 2022-11-25

**Authors:** Liming Luo, Huiyun Peng, Hongjuan Sun, Tongjiang Peng, Mingliang Yuan

**Affiliations:** 1Fundamental Science on Nuclear Wastes and Environmental Safety Laboratory, Southwest University of Science and Technology, Mianyang 621010, China; 2Institute of Mineral Materials & Application, Southwest University of Science and Technology, Mianyang 621010, China; 3School of Mineral Processing and Bioengineering, Central South University, Changsha 410083, China

**Keywords:** nano-assembly, α-MnO_2_/reduced graphene oxides, three-dimensional structure, pore diameter distribution, super-capacitive performance

## Abstract

A series of three-dimensional porous composite α-MnO_2_/reduced graphene oxides (α-MnO_2_/RGO) were prepared by nano-assembly in a hydrothermal environment at pH 9.0–13.0 using graphene oxide as the precursor, KMnO_4_ and MnCl_2_ as the manganese sources and F^−^ as the control agent of the α-MnO_2_ crystal form. The α-MnO_2_/RGO composites prepared at different hydrothermal pH levels presented porous network structures but there were significant differences in these structures. The special pore structure promoted the migration of ions in the electrolyte in the electrode material, and the larger specific surface area promoted the contact between the electrode material and the electrolyte ions. The introduction of graphene solved the problem of poor conductivity of MnO_2_, facilitated the rapid transfer of electrons, and significantly improved the electrochemical performance of materials. When the pH was 12.0, the specific surface area of the 3D porous composite material αMGs-12.0 was 264 m^2^·g^−1^, and it displayed the best super-capacitive performance; in Na_2_SO_4_ solution with 1.0 mol·L^−1^ electrolyte, the specific capacitance was 504 F·g^−1^ when the current density was 0.5 A·g^−1^ and the specific capacitance retention rate after 5000 cycles was 88.27%, showing that the composite had excellent electrochemical performance.

## 1. Introduction

Manganese dioxide (MnO_2_) has a wide potential window and outstanding electrochemical properties and it has been widely applied in electrode materials. The various manganese–oxygen octahedron connection modes mean that MnO_2_ has multiple crystal structures and unique physicochemical properties. As the nanofiber manganite with a tunnel structure, α-MnO_2_ has a large specific surface area [1,2], high porosity [3,4] and high oxygen vacancy density [5,6]. The unique crystal form and structural features are the reason behind its outstanding electrochemical properties. If α-MnO_2_ and graphene were to be assembled into three-dimensional porous composites at the nanoscale, and the variation of structure, morphology, pore diameter distributions and electrochemical properties investigated, this would be of great theoretical significance for the design and development of new composite electrode materials.

α-MnO_2_ has attracted attention in the fields of electrode materials, adsorption materials and catalysis materials [7] because of its outstanding physicochemical properties. Since the development of the Leclanche battery with a MnO_2_ cathode [8], MnO_2_ electrode materials have been discussed extensively, and α-MnO_2_ shows even better electrochemical performance. Previous studies on α-MnO_2_ mainly focused on its tunnel structure [9], energy bands [10], surface defects [11,12], porous structure [13] and charge–discharge mechanism [14,15]. Recently, there has been research into the electrochemical performance of α-MnO_2_ due to the growth of new energy development and storage system demands. These studies have focused on the material structure, morphology, adulteration and composition to determine ways to effectively improve the specific capacitance, ratio discharge property and cycling stability. For instance, secondary wall-like structure α-MnO_2_ nanospheres were prepared by constant current cathodic deposition [16], and α-MnO_2_ nanospheres were synthesized by reflux and DC magnetron sputtering technology [17,18] showing a specific capacitance of 346 F·g^−1^ (0.71 A·g^−1^). In other studies, Al [19], Co [20], Ce^3+^ [21], Ni [22] and Fe [23] were mixed into α-MnO_2_ producing materials with high specific capacitance (200–650 F·g^−1^). Subsequently, excellent composites (e.g., α-MnO_2_/3DGF [24,25], α-MnO_2_/CFP [26] and α-MnO_2_/GR/CNT [27]) have been prepared by combining α-MnO_2_ and carbon materials in order to further increase the specific capacitance and cyclic stability of the electrode material.

Similarly, as a ‘hot spot’ for carbon material research, reduced graphene oxide (RGO), which is combined with MnO_2_ to prepare electrode materials, has also attracted much attention. MnO_2_/RGO electrode materials have been prepared by the manganese powder reduction method [28], sonochemical method [29], reverse microemulsion method [30], reduction-induced in situ self-assembly method [31] and one-step hydrothermal synthesis method [32]; these MnO_2_/RGOs had high specific capacitance and good cycle stability. However, in order to further improve the electrochemical performance of MnO_2_/RGOs, researchers prepared MnO_2_/NRGO [33], Ag/MnO_2_/RGO [34], PPy/MnO_2_/RGO [35] and MnO_2_/RGO@Ni-foam [36] by doping or compounding with N, Ag, PPy, Ni, respectively; these materials exhibited superior supercapacitor performance. Although there have been many studies on the three-dimensional porous structure of α-MnO_2_/RGOs, no in-depth study has been conducted on effects of the controllable aspects of the preparation methods at the nanometer scale, especially on the three-dimensional structure, defects, pore distribution and other changes in the composite process.

Based on previous studies on the preparation, structure and physicochemical properties of GOs and α-MnO_2_, in view of the shortcomings of MnO_2_ such as poor conductivity, small specific surface area, low diffusion/transmission efficiency of electrons and ions in electrode materials, and insufficient redox reaction, in this paper, α-MnO_2_ was cultured on the surface of GOs and self-assembled into a 3D composite by controlling the pH in a hydrothermal environment. Structure, microstructure, specific surface area, pore diameter distribution and super-capacitive performance were characterized and analyzed by X-ray diffraction (XRD), X-ray photoelectron spectroscopy (XPS), Raman spectroscopy, scanning electron microscopy (SEM), energy dispersive spectroscopy (EDS), specific surface area analysis (BET), pore diameter analysis (BJH) and an electrochemical workstation. This paper describes the influence of hydrothermal conditions on the structure, physicochemical properties and electrochemical properties of the prepared composites. This work achieved the complementary advantages of MnO_2_ and graphene, and the prepared composite material has high specific capacitance and cycle stability, which can effectively promote the application of MnO_2_ composite electrode material in supercapacitors.

## 2. Experimental

### 2.1. Raw Materials and Reagents

Graphene oxide sample (GO-60) was prepared by the improved Hummers method [37]. The mass ratio of graphite and KMnO_4_ was 1:4.

Analytically pure KMnO_4_, MnCl_2_·4H_2_O, NaF and 25–28% NH_3_·H_2_O were purchased from Chengdu Kelong Chemical Reagent Factory, China. Ultrapure water with specific resistance ≥18.25 MΩ·cm was used throughout.

### 2.2. Preparation of Samples

Preparation of the KMnO_4_-GO mixed dispersion solution: GO gel was added to deionized water (50 mL, 4.0 mg/mL). Ultrasonic treatment was performed for 2 h to produce the GO dispersion liquid. Next, KMnO_4_ (0.026 g) was added and dissolved completely.

Preparation of MnCl_2_-NaF solution: MnCl_2_·4H_2_O (0.049 g) was added to deionized water (50 mL) and dissolved completely. Next, NaF (0.021 g, crystal form control agent) was added and dissolved completely.

Preparation of α-MnO_2_/RGO: KMnO_4_-GO mixed dispersion solution (50 mL) was dropped gradually into a beaker containing the MnCl_2_-NaF solution (50 mL) and NH_3_·H_2_O was added to adjust the pH. The mixture was stirred for 30 min and the composite mixture (n_KMnO4_:n_MnCl2_:n_NaF_ = 1:1.5:3, GO content = 2.0 mg/mL) was transferred into a 150 mL Teflon reaction tank. The α-MnO_2_/RGO 3D composite was prepared after 8 h hydrothermal treatment at 180 °C. According to the different hydrothermal pH levels, the samples are referred to as αMGs-X, where X = 9.0, 10.0, 11.0, 12.0 and 13.0.

### 2.3. Sample Analysis

For X-ray diffraction, an X’pert MPD Pro-type X-ray diffractometer was used with Cu Ka radiation (λ = 0.15406 nm) operated at 40 kV and 40 mA (XRD, PANalytical, Almelo, The Netherlands). For Raman spectral analysis, an InVia laser Raman spectrometer was used (Raman, Renishaw, London, UK). For Morphological analysis image, an Ultra55 scanning electron microscope was used (SEM, Zeiss, Oberkochen, Germany). For X-ray photoelectron spectroscopy, an XSAM 800 multi-function surface analysis electron spectrometer (XPS, Kratos, Manchester, UK) was used. For Elemental composition, an Oxford IE450X-Max80 EDX spectrometer was used, operated at 15 kV with spot size of 2 nm (EDS, Zeiss, Oberkochen, Germany). For surface area and pore diameter analysis, an Autosorb-1MP equipment was used (BET and BJH, Quantachrome, Boynton Beach, FL, USA). For the super-capacitive performance, a Shanghai Chenghua Instrument (Shanghai, China) was used. Na_2_SO_4_ solution (1.0 mol·L^−1^) was used as the electrolyte and platinum sheets (1.5 × 1.5 cm) were used as the counter electrodes. A Hg/HgO electrode was used as the reference electrode. The three-electrode test system was chosen.

## 3. Results and Discussion

### 3.1. Morphology, Microstructure, and Zeta Potential

The appearance and microstructure of the αMGs-X samples prepared at different hydrothermal pH levels are shown in Figure 1. All samples formed a stable 3D column structure. However, with the increase in pH, the 3D column structural volume initially increased and then decreased. According to microstructure characterization, all αMGs-X samples presented a porous network structure under microscopic analysis. However, the pore diameters and morphologies were different at different hydrothermal pH levels. At pH 9.0, the pore diameter of the 3D composite was relatively large (≥5 μm) and the cluster structure was formed of RGO sheets with large gaps. At pH 10.0, the pore diameter of the composite decreased (≥2 μm) and the RGO sheets overlapped into a porous structure. When the pH was 11.0 or 12.0, the composite developed a uniform superfine cellular porous structure (≤1 μm) and the sample volume was the largest. At pH 13.0, the pore diameter of the composite was extremely large (≥8 μm) but the volume decreased.

The zeta potential of the GO dispersion solution was determined before the hydrothermal reaction (Figure 2a) and was negative when the pH was between 9.0 and 13.0. However, the absolute value of the zeta potential of the GO solution initially increased and then decreased with increasing pH: the zeta potential values were −41.35, −60.46 and −50.43 mV at pH 9.0, 12.0 and 13.0, respectively. According to this analysis, the absolute value of the zeta potential of the GO solution was relatively small when the hydrothermal pH was low and accompanied by a weak electrostatic repulsive force between the GO sheets and poor dispersion. Most oxygen-containing functional groups were eliminated during the hydrothermal process and RGO were formed. Van Der Waals’ forces resulted in overlap of the RGO layers. Although the pore diameter was large, the composite volume was small, which was attributed to the piling of RGO. When the pH was too high, the zeta potential of the GO solution was high, but the number of oxygen-containing functional groups on the GO surface decreased because some were neutralized by the alkaline solution. As a result, some GOs were reduced and reunited, which decreased the volume after hydrothermal treatment. Moreover, there was abundant NH_4_^+^ in the high-concentration ammonium hydroxide, which reacted with oxidized graphene to form N-adulterated RGOs. N-adulterated RGOs have a large diameter [38] and the pore diameter of the resulting composite was large. When the pH was 11.0 and 12.0, the absolute value of the zeta potential of the GO was at a maximum. In the solution, there was a high anion concentration and the GO surface carried abundant negative charges; the strong electrostatic repulsive force between the GO sheets resulted in more even dispersal of the layers. In addition, there were strong hydrogen bond interactions between the COO^−^ and water molecules in the alkaline environment. The GOs were highly hydrophilic and the RGOs formed by hydrothermal reduction covered the fiber-like α-MnO_2_ surface, thus forming the stable porous network structure with a uniform distribution of pore diameters.

Comparison of the solution pH before and after the hydrothermal reaction (Figure 2b) showed that, when the hydrothermal pH was low (9.0 and 10.0), the solution pH before and after the hydrothermal reaction changed greatly and even became acidic. When the pH was high (11.0, 12.0, and 13.0), the pH changed only slightly and was alkaline. This was mainly due to two reasons: firstly, volatilization of some NH_3_ decreased the alkalinity of the solution in the hydrothermal process, thus decreasing the pH; secondly, due to the presence of F^−^, the solution produced a large amount of hydrogen ions during the hydrothermal process. Hence, the pH of samples dropped sharply to pH 2.57.

### 3.2. Microstructure and Structural Characterization

High-magnification SEM analysis of αMGs-X samples was carried out to further analyze the combination of α-MnO_2_ and graphene, as well as the form of the combined α-MnO_2_. All αMGs-X samples presented a porous structure, but samples showed different high-magnification morphologies at different hydrothermal pH levels. When pH was 9.0, 10.0 and 13.0 (Figure 3a,b,f, respectively), no α-MnO_2_ nanorods were seen in the high-magnification morphology, which might have been due to their envelopment in RGO. When the pH was 11.0 (Figure 3c) and 12.0 (Figure 3d), a 3D porous network structure formed by RGO was observed from low-magnification SEM. High-magnification SEM pictures showed that each hole was formed by the interaction of α-MnO_2_ nanorods (Figure 3e), and these α-MnO_2_ nanorods were dispersed evenly and had a uniform size. These data confirmed that the experiment resulted in a perfect nanoscale combination of α-MnO_2_ nanorods and graphene.

It is necessary to analyze the distribution of C and Mn in the materials to determine the distribution of α-MnO_2_ in the 3D composites. Analysis of the elemental distribution of the αMGs-12.0 sample showed it to contain C and Mn (Figure 4). The Mn distribution was consistent with the cross-linking shapes of α-MnO_2_ in the 3D graphene holes, which supports the data (shown in Figure 3c,d) that the 3D porous structure was formed by α-MnO_2_ and graphene. The proportions of Mn and C in αMGs-12.0 were C/Mn = 0.7:1, which reflects the relatively high Mn content in the 3D composites.

As shown in Figure 5, XRD analysis of samples revealed that α-MnO_2_ developed three characteristic diffraction peaks at 2θ = 28.712°, 37.536° and 49.835°, which corresponded to (310), (211), and (411) surfaces of α-MnO_2_ (JCPDS 44-0141). After hydrothermal treatment, none of the αMGs-X samples developed a diffraction peak characteristic of α-MnO_2_. In addition, αMGs-X samples developed a characteristically wide diffraction peak close to 20–25° (d = 0.37 nm) [39]; the intensity of this peak initially decreased and then increased slightly with increasing pH. According to analysis, this was the characteristic diffraction peak of graphite oxide-like substances (d_002_ = 3.35 Å for graphite) [37], which indicated cohesion of the RGO sheets in the samples and the formation of an ordered stratified structural superposition.

When the hydrothermal pH was alkaline, GOs were poorly dispersed and there was considerable piling of RGOs in the composites after the hydrothermal process. The degree of piling order was high, as was the formed diffraction peak intensity. As the pH increased, GO dispersion improved but piling of the RGOs was difficult after the hydrothermal process and the formed diffraction peak intensity was low. At pH 13.0, the alkalinity was high and GO was chemical-reduced, accompanied by agglomeration. RGOs piled up in composites after the hydrothermal process, which resulted in a high intensity of the characteristic diffraction peak.

Raman characterization of the αMGs-X samples prepared at different hydrothermal pH levels was used to determine the reduction of GO to RGO in the hydrothermal process (Figure 6a). All αMGs-X samples developed D, G, 2D, and D+G Raman peaks characteristic of RGO at wave numbers 1352, 1589, 2716, and 2940 cm^−1^, respectively. The value of the integral intensity ratio (I_D_/I_G_) of peaks D and G and the mean of the relative size L_a_ of the graphene microcrystalline structure formed by sp^2^ hybridization of carbon atoms were used to characterize the degree of structure disorder and defect changes of RGO after the hydrothermal process (Figure 6b) [40].

In Figure 6, the I_D_/I_G_ of the αMGs-X samples was positively related to the hydrothermal pH while L_a_ was negatively correlated, which indicated that new defects were produced by the reduction in GO during the hydrothermal process. Consequently, the region of graphite microcrystalline structure, which was formed by sp^2^ hybridized carbon atoms, was narrowed, marginal defects increased, and the degree of disorder of the αMGs-X samples increased. At pH 13.0, the I_D_/I_G_ of the αMGs-13.0 sample decreased and L_a_ increased, which reflected that GO was chemical-reduced, there were few defects and a large zone of microcrystalline graphite due to the simultaneous action of N adulteration and the hydrothermal process. These data confirmed the high-magnification SEM analysis results in Figure 3e.

In order to analyze the chemical composition and elemental state of the three-dimensional composite sample, XPS characterization analysis was performed on the GO-60 and αMGs-12.0 samples (Figure 7). It can be seen from the full XPS spectrum (Figure 7a) that the characteristic peaks of C1s and O1s appeared in the GO-60 sample and the peak intensity of O1s was higher than that of C1s. The sample αMGs-12.0 also showed characteristic peaks of C1s and O1s, but the intensity of the O1s peak was significantly reduced; this related to the reduction of oxygen-containing functional groups during the hydrothermal process. In addition, αMGs-12.0 also showed characteristic peaks of Mn (2p_3/2_, 2p_1/2_) with high intensity, and characteristic peaks of N1s with weaker intensity, indicating that the composite sample had a higher Mn content but that the RGO was doped by N during the hydrothermal process.

Figure 7b shows the C1s spectrum of the GO-60 and αMGs-12.0 sample. The peaks at 284.5, 286.4, 287.5 and 288.5 eV were attributed to the characteristic peaks of C=C, C−OH, C−O−C and O−C=O, respectively [41,42]. After hydrothermal recombination, the content of C−OH and C−O−C in the composite were significantly decreased, and the characteristic peak of O−C=O disappeared, indicating that most of the GO had been reduced and a small amount of C-OH and C−O−C remained on the surface. In Figure 7c, 641.9 eV± corresponds to the electron binding energy of Mn2p_3/2_, and 653.5 eV± corresponds to the electron binding energy of Mn2p_1/2_ [43], indicating the presence of Mn in the three-dimensional composite sample in two valence states: Mn^4+^ (641.9 eV) and Mn^3+^ (653.5 eV).

Figure 7d shows the XPS spectrum of N1s. N1s was divided into two characteristic peaks at 399.7 eV and 401.1 eV, corresponding to the graphite nitrogen and nitrogen oxide in the sample. This indicated that in the hydrothermal process, the elemental N in solution was doped with RGO. By calculating the integral areas of the characteristic peaks of C1s, Mn, N1s and O1s on the surface of the αMGs-12.0 sample, the percentages of C, Mn, N and O on the surface of the sample were determined as 38%, 50%, 3% and 9%, respectively.

### 3.3. Specific Surface Area and Pore Diameter Distribution

To verify the porous structure of the 3D composites, a low-temperature nitrogen adsorption–desorption test of the αMGs-12.0 sample was carried out (Figure 8). The αMGs-12.0 sample had the most uniform porous structure and pore diameter distribution. Figure 8a shows that the nitrogen adsorption–desorption curve of the sample at low temperature was a typical IV-type curve and had a hysteresis loop in the relative pressure range of 0.6–1.0. Moreover, the specific surface area of the αMGs-12.0 sample reached 264 m^2^·g^−1^ according to the BET calculation. Hence, the αMGs-12.0 composite had a typical porous structure; this could effectively relieve reunite of α-MnO_2_, while the large specific surface area of the composite provided a larger space for ion storage in the electrolyte environment.

The porous structure of the electrode material could provide a good transmission and migration channel for ions in the electrolyte. Generally, the pore diameter of the electrode material was 3–50 nm, which was conducive to improving the capacitive performance of materials. The pore diameter distribution of the αMGs-12.0 sample based on the BJH calculation is shown in Figure 8b; this sample had a wide pore diameter distribution interval (20–40 nm) that was mainly concentrated at about 25 nm. The pore diameter distribution of the samples was relatively uniform (consistent with the SEM analysis in Figure 3d), so it was beneficial to improving the super-capacitive performance.

### 3.4. Electrochemical Properties

The cyclic voltammetry (CV) curve, charge–discharge (GCD) curve under a current density of 0.5 A·g^−1^, and the specific capacitance at different current densities of the αMGs-X samples are shown in Figure 9a–c, respectively. The scanning rate was fixed at 5 mV·s^−1^. The relationships between the specific capacitance of αMGs-12.0 and α-MnO_2_ with the number of cycles at a current density of 0.5 A·g^−1^ are shown in Figure 9d.

In Figure 9a, the CV curves of the αMGs-X samples prepared under different hydrothermal pH levels all showed similar rectangular shapes and had a wide redox peak. The potential interval was 1.0 V. This redox peak may be attributed to the pseudocapacitance effect caused by α-MnO_2_ in materials. The GCD curves of all αMGs-X samples in Figure 9b also show similar isosceles triangle shapes. The αMGs-12.0 sample had the longest discharge time, followed by αMGs-11.0, αMGs-10.0, αMGs-9.0 and αMGs-13.0, successively. All samples showed evident characteristics of electrical double-layer capacitance.

The specific capacitance of different αMGs-X samples can be calculated from the formula C_GCD_ = I·∆t/(m·∆V). The relationships between the specific capacitance of the αMGs-X samples and current density are shown in Figure 9c. With increased current density, the specific capacitance of all samples decreased gradually. At a current density of 0.5 A·g^−1^, the specific capacitance of the αMGs-X sample initially increased and then decreased with increasing hydrothermal pH. Specifically, αMGs-12.0 achieved the highest specific capacitance (504 F·g^−1^) when the hydrothermal pH was 12.0. When the current density was high (5.0 A·g^−1^), all samples maintained high specific capacitance, indicating the good rate capability of samples.

In Figure 9d, the attenuation amplitude of specific capacitance was large in the first 1000 cycles but then decreased dramatically in the last 4000 cycles. This was mainly because in the first 1000 charge–discharge cycles, some manganese ions from α-MnO_2_ in the composites entered the electrolyte rather than providing pseudocapacitance. Hence, the overall specific capacitance dropped sharply. After 5000 charge–discharge cycles, the specific capacitance of the materials dropped from 504 F·g^−1^ to 445 F·g^−1^, and the capacity retention rate reached 88.27%. The material showed excellent cyclic stability. The specific capacitance of the composite materials made of different carbon materials and MnO_2_ is shown in Table 1. Compared with the specific capacitance determined by previous studies, the results in this paper are higher than those of composites of other carbon materials and MnO_2_.

## 4. Conclusions

Composite self-assembly of materials on the nanoscale was accomplished by the hydrothermal process to prepare α-MnO_2_/RGO 3D porous composites. Structure, morphology and super-capacitive performance of αMGs-X were systematically analyzed. The morphology and porous structure of αMGs-X can be controlled effectively by adjusting the hydrothermal pH. αMGs-12.0 had an extremely good porous network structure and the porous structure was mainly formed by the interaction of α-MnO_2_ nanorods and RGO. The α-MnO_2_ nanorods were uniformly dispersed and the mean pore diameter was approximately 25 nm. The excellent porous structure could provide a good channel for ion transmission and migration, which was conducive to improving the electrochemical properties of the materials. To summarize, hydrothermal nano-assembly was a direct and effective method to produce a nanoscale material composition and has promising application prospects in the development of MnO_2_-based composite electrode materials and advanced super-capacitors.

## Figures and Tables

**Figure 1 materials-15-08406-f001:**
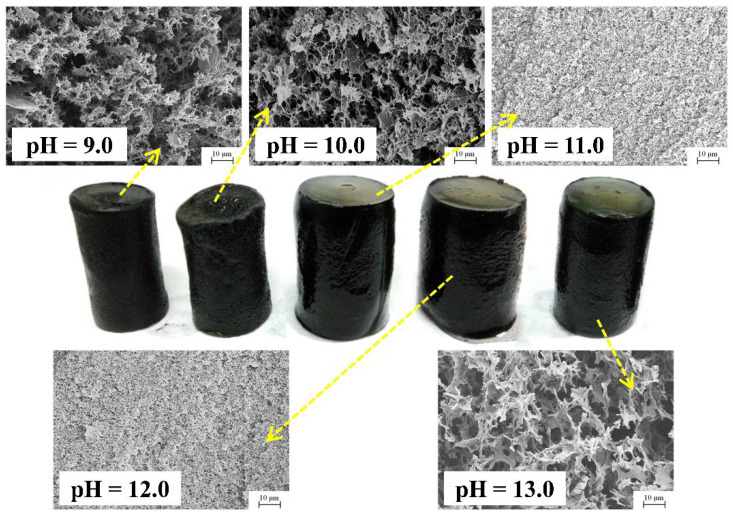
Appearance and microstructure of αMGs-X samples prepared at different pH levels.

**Figure 2 materials-15-08406-f002:**
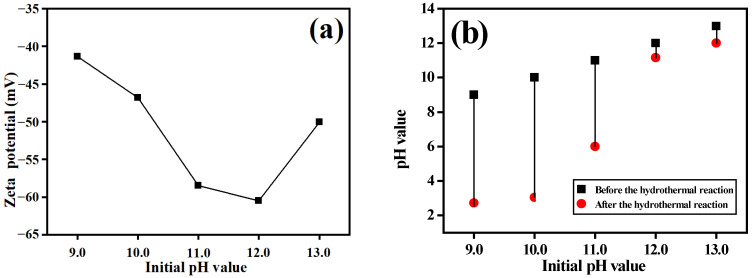
(**a**) Zeta potential at different pH levels. (**b**) pH changes of the solution before and after hydrothermal treatment.

**Figure 3 materials-15-08406-f003:**
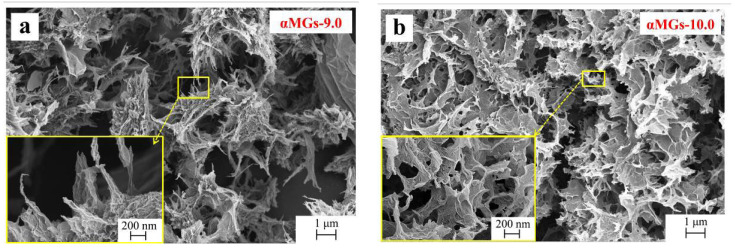
High-magnification SEM images of αMGs-X samples at different hydrothermal pH levels.

**Figure 4 materials-15-08406-f004:**
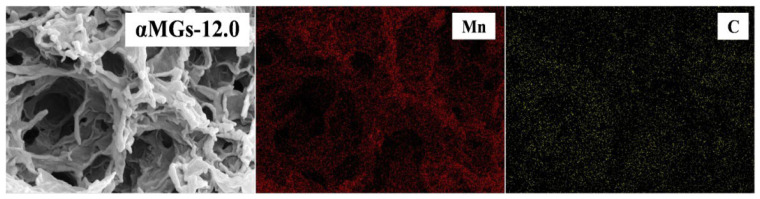
Elemental distributions in αMGs-12.0.

**Figure 5 materials-15-08406-f005:**
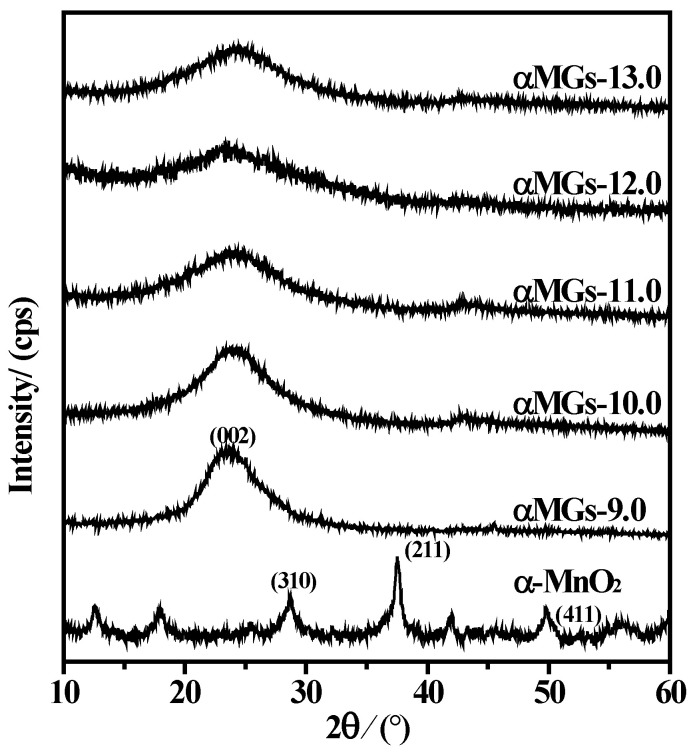
XRD patterns of αMGs-X samples at different hydrothermal pH levels.

**Figure 6 materials-15-08406-f006:**
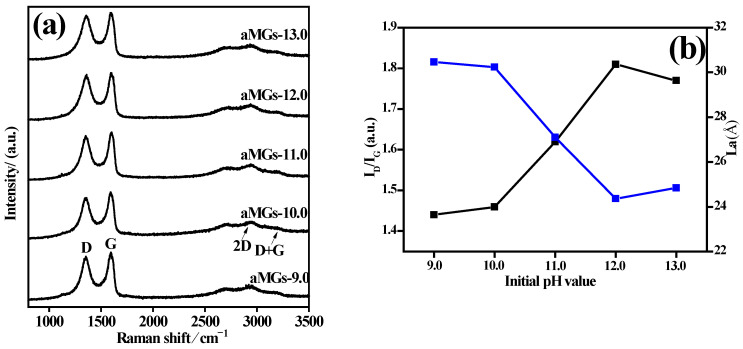
(**a**) The Raman spectra; (**b**) I_D_/I_G_ (black) and L_a_ (blue) of αMGs-X samples.

**Figure 7 materials-15-08406-f007:**
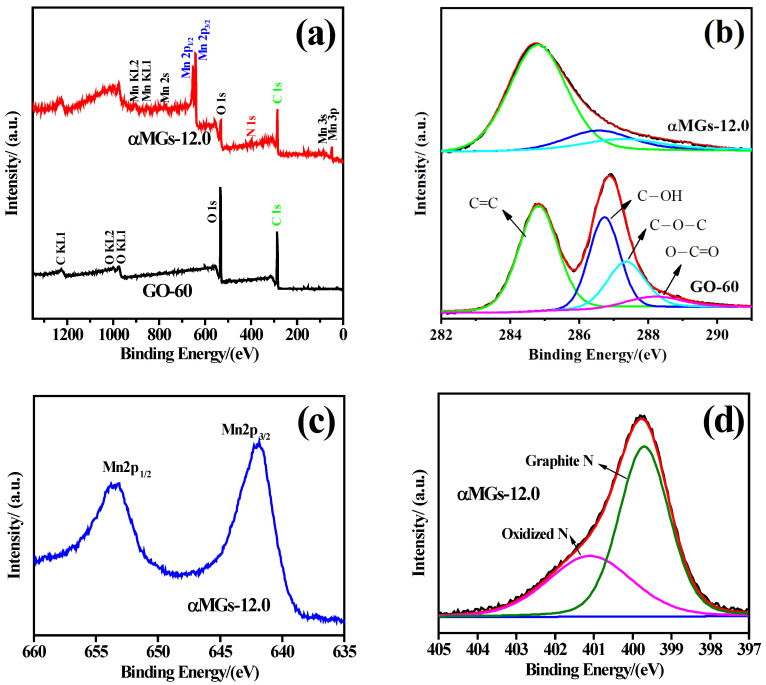
XPS spectra of GO-60 and αMGs-12.0 sample. (**a**) Full XPS spectrum; (**b**) C1s spectrum; (**c**) Binding energy of manganese; (**d**) XPS spectrum of N1s.

**Figure 8 materials-15-08406-f008:**
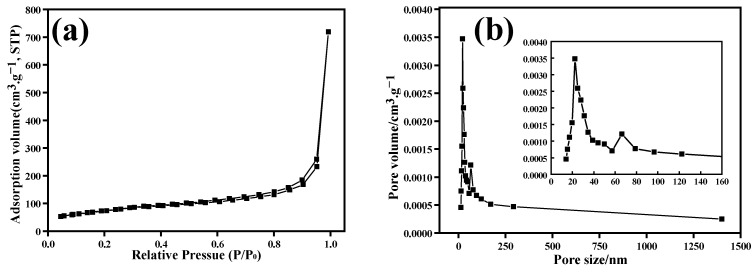
(**a**) N_2_ adsorption–desorption isotherms. (**b**) Pore size distributions.

**Figure 9 materials-15-08406-f009:**
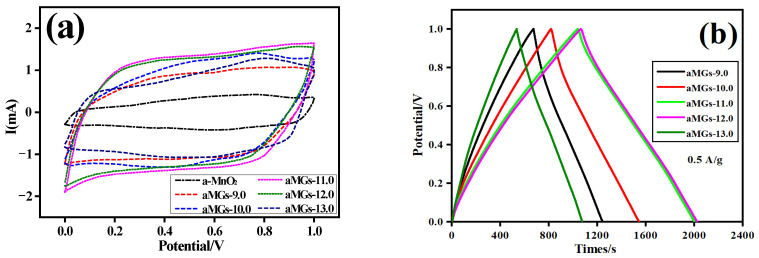
(**a**) CV curve of αMGs-X samples at a scanning rate of 5 mV·s^−1^. (**b**) GCD curves at a current density of 0.5 A·g^−1^. (**c**) Specific capacitance at different current densities. (**d**) Relationship between specific capacitance of αMGs-12.0 and α-MnO_2_ and number of cycles at a current density of 0.5 A·g^−1^.

**Table 1 materials-15-08406-t001:** The capacitance of MnO_2_–carbon composites.

Sample Name	Capacitance	Reference
Mesoporous MnO_2_/PPy nanofilms	320 F·g^−1^ at 0.5 A·g^−1^	[44]
Graphene/MnO_2_ nanoflower	321 F·g^−1^ at 0.5 A·g^−1^	[45]
3D graphene/carbon nanotubes/MnO_2_ nanoneedles	343 F·g^−1^ at 2 mV·s^−1^	[46]
Carbon fibers sheet/MnO_2_	375 F·g^−1^ at 0.5 A·g^−1^	[47]
PPy nanotube/MnO_2_	403 F·g^−1^ at 0.5 A·g^−1^	[48]
Electrochemical reduced GO/MnO_2_	423 F·g^−1^ at 0.5 A·g^−1^	[49]
FGS-SSG(Graphene)/MnO_2_	465 F·g^−1^ at 2 mV·s^−1^	[50]
Carbon fiber fabric/MnO_2_ composites	467 F·g^−1^ at 0.5 A·g^−1^	[51]
3D graphene/α-MnO_2_	504 F·g^−1^ at 0.5 A·g^−1^	This work

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
