# Peer review of "Research on Three-Dimensional Porous Composite Nano-Assembled α-MnO2/Reduced Graphene Oxides and Their Super-Capacitive Performance"

_materials, 2022, doi:10.3390/ma15238406_

Round 1

Reviewer 1 Report

The manuscript is devoted to synthesis and characterization of novel material – composite of α-MnO2 with reduced graphene oxide. The authors found the protocol for hydrothermal treatment of this composite which leads to highly porous structure. This material demonstrates excellent performance in supercapacitors, both in terms of capacitance and stability. Thus, this paper deserves to be published.

However, several drawbacks (mainly typos and incomplete description of experimental results) should be corrected:

1.     Paragraph 2.3, line 4. The word “photographs” should be corrected to “image”. This is because SEM image is generated not by photons, but by electrons.

2.     Paragraph 2.3, lines 5-6. The sentence “X-ray photoelectron spectroscopy used an XSAM 800 multi-function surface analysis electron spectrometer” should be re-phrased in passive voice: “For X-ray photoelectron spectroscopy an XSAM 800 multi-function surface analysis electron spectrometer was used”. Similar corrections should be applied for several further sentences.

3.     Paragraph 2.3, line 4. The word “change” should be corrected to “increase” to be more precise.

4.     In the caption of Fig. 6 it should be explained what is shown in each of the panels (A) and (b). For Fig. 6b it should also be indicated for which axis (left or right) the black and the blue symbols correspond.

5.     The top right panel of Fig. 7 should be marked as (b), not (c).

6.     The top right panel of Fig. 9 should be marked as (b).

Reviewer 2 Report

The present paper entitled " Research on Three-dimensional Porous Composite Nano-assembled α-MnO2/Reduced graphene oxides and their Super-capacitive Performance" by Luo et al. presents the series of three-dimensional porous composite α-MnO2/reduced graphene oxides (α-MnO2/RGO) which were prepared by nano-assembly in a hydrothermal environment at pH 9.0–13.0 using graphene oxide as the precursor, KMnO4 and MnCl2 as the manganese sources and F- as the control agent of the α-MnO2 crystal form and shown that the composite had excellent electrochemical performance. The work is nice but I have some major concern given below.

My detail comments are following:

1.     Aim of the paper is not clear. Author should explain the relevance of the study and also include importance of the work more briefly in terms of application point of view both in manuscript as well as in abstract.

2.     There are some previous studies like “Layer-by-Layer Heterostructure of MnO2@Reduced Graphene Oxide Composites as High–Performance Electrodes for Supercapacitors”, “Facile Fabrication of MnO2/Graphene/Ni Foam Composites for High-Performance Supercapacitors” and “Three-dimensionally assembled Graphene/α-MnO2 nanowire hybrid hydrogels for high performance supercapacitors”. Author should explain the difference of there work with the previously reported work.

Sorry but I do not recommend this manuscript for publication in Materials in the current form. Throughout this paper a more detailed comparison to the previous work is needed and a justification of what is really new here.

Reviewer 3 Report

The authors in the present manuscript to show that a series of three-dimensional porous composite α-MnO2/reduced graphene oxides (α-MnO2/RGO) were prepared by nano-assembly in a hydrothermal environment at pH 9.0–13.0 using graphene oxide as the precursor, KMnO4 and MnCl2 as the manganese sources and F- as the control agent of the α-MnO2 crystal form. The α-MnO2/RGO composites prepared at different hydrothermal pH levels presented porous network structures but there were significant differences in these structures. When the pH was 12.0, the specific surface area of the 3D porous composite material αMGs-12.0 was 264 m2·g-1, and it displayed the best super-capacitive performance: in Na2SO4 solution with 1.0 mol∙L-1 electrolyte, the specific capacitance was 504 F∙g-1 when the current density was 0.5 A∙g-1 and the specific capacitance retention rate after 5000 cycles was 88.27%, showing that the composite had excellent electrochemical performance. The authors should address the following issues and information’s before publication acceptance in the prestigious ‘Materials’ Journal:

1. In Introduction, authors should incorporate a Table for comparison of specific capacitance, cycles and properties of your study with published papers on similar works.

2. Authors should include materials preparation schematic flow diagram to help better understand.   

3. In 2.3. Sample analysis, Page 3, authors have repeated two times ‘specific surface area and pore diameter’ words. Authors may mention the relative pressure (P/Po) to calculate surface area?

4. In 3.1. Morphology, microstructure, and Zeta potential, Page 3, authors should add the details how to calculate pore diameter of samples using SEM images and also add the SEM image of RGO for better comparison.  

5. In Figure 5, Page 7, authors can explain that why sample at low pH = 9 shows higher crystalline nature of graphite peak compared to samples at higher pH = 10-13. Add one more reference to show graphitic peak in the range of 20-25°. Authors may incorporate this reference: Materials Chemistry and Physics 239 (2020) 122102.

6. In Page 7, authors should add the references related to D, G, 2D, and D+G for Raman analysis.  

7. In Figure 7, authors can mention the details of used software for peak fitting of C1s, O1s and N1s and also incorporate the references for functional groups in the text. Authors may refer these papers: Journal of Industrial and Engineering Chemistry 20 (2014) 2883–2887, RSC Adv., 2016, 6, 106914 and Journal of Porous Materials (2021) 28:875–888.

8. Author should explain in electrochemical section about other properties (functional groups, graphitic nature, etc.) to improve or correlates to super-capacitive performance than surface area.

Round 2

Reviewer 2 Report

Auther have done significant improvements as suggested. I can recommend the manuscript for publication.